# Single-positive Multi-label Learning with Label Cardinality

**Shayan Gharib**                                                 *shayan.gharib@helsinki.fi*
*Department of Computer Science*
*University of Helsinki*

**Pierre-Alexandre Murena**                        *pierre-alexandre.murena@tuhh.de*
*Human-Centric Machine Learning Research Group*
*Hamburg University of Technology*

**Arto Klami**                                                        *arto.klami@helsinki.fi*
*Department of Computer Science*
*University of Helsinki*

**Reviewed on OpenReview:** *https://openreview.net/forum?id=XEPPXH2nKu*

## Abstract

We study learning a multi-label classifier from partially labeled data, where each instance has only a single positive label. We explain how auxiliary information available on the label cardinality, the number of positive labels per instance, can be used for improving such methods. We consider auxiliary information of varying granularity, ranging from knowing just the maximum number of labels over all instances to knowledge on the distribution of label cardinalities and even the exact cardinality of each instance. We introduce methods leveraging the different types of auxiliary information, study how close to the fully labeled accuracy we can get under different scenarios, and show that an easy-to-implement method only assuming the knowledge of the maximum cardinality is comparable to the state-of-the-art single-positive multi-label learning methods when using the same base model. Our implementation is publicly available at `https://github.com/shayangharib/SPMLL_with_Label_Cardinality`.

## 1 Introduction

Numerous methods for learning multi-label classifiers from fully annotated data have been proposed (Ridnik et al., 2023; Zhang & Zhou, 2013; Parascandolo et al., 2016), but for specialized domains full annotation is often too costly (Nguyen et al., 2017; Xie & Huang, 2018). Instead, we aim to train a model with minimal label information – ideally, just a single positive label – which is far less costly than annotating all possible aspects. Algorithms for learning the full multi-label function from such labeling are commonly called single-positive multi-label learning (SPMLL) (Cole et al., 2021), which is a special case of the more general positive and unlabeled (PU) setting (Elkan & Noto, 2008).

The SPMLL literature has largely focused on the question of how the missing label information is treated, by introducing new losses (Chen et al., 2024; Kim et al., 2022), regularizers (Cole et al., 2021) or algorithmic variants (Xie et al., 2022). We, instead, focus on leveraging a new type of complementary information: How additional information about *label cardinality* can be used for improving SPMLL methods. Here label cardinality refers to the number of positive labels of an instance. Direct access to the cardinality of a specific instance is naturally highly useful – for instance, knowing an instance has just one label implies that all unobserved labels are negative – but it is often difficult to obtain in practice. We explain and evaluate alternative ways of using label cardinality to improve SPMLL methods and quantify the possible gains that can be achieved in different setups. Our main focus is in cases where we assume known *cardinality distribution*, a probability distribution over the label cardinalities, and use this information to estimate

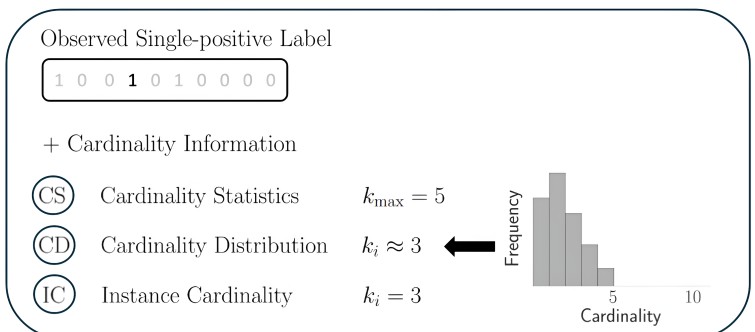

Figure 1: In SPMLL, only one positive label (in black) for each instance is observed, yet the task is to predict all labels. We explain how to use different forms of additional information on label cardinality, the number of positive labels, to help this. The information can be on data-level cardinality statistics (CS), instance-level cardinality (IC), or cardinality distribution (CD) over all samples.

instance-level cardinalities. This is done with a newly proposed algorithm that finds the optimal cardinalities based on the current model predictions, without requiring label information for learning. However, we also study the case where the cardinalities are known exactly, and a setup where only crude summary statistics of the cardinality distribution are observed. Figure 1 illustrates the setup.

Several studies have explored the use of label cardinalities in other contexts. For instance, label cardinality has been used as a proxy for scene complexity (Hajimirsadeghi et al., 2015), cardinality constraints can be used to enforce models to output a predefined number of labels (Cortes et al., 2024; Amos et al., 2019), and the related concept of cardinality potentials has been considered in count regression problems to encourage predictions to be on the right scale (Swersky et al., 2012; Tarlow et al., 2012). However, within SPMLL literature this aspect has been ignored, even though additional information is most useful when having the least amount of direct label information. The only previous reference to label cardinality is by Cole et al. (2021), who used the expected number of positive labels on the level of the whole dataset as a basis for a regularizer. This can be seen as the crudest special case of our principle.

On a technical level, we formalize the task by characterizing the different forms of cardinality side information that could be available, and provide concrete solutions for leveraging them. We introduce three key components and a specific method combining them, but the components could be incorporated into various other SPMLL methods as well.

1. A new loss that ensures the outputs for the unobserved classes remain informative during training, in contrast to the previous losses pushing them towards arbitrary target values (Cole et al., 2021; Zhou et al., 2022).

2. A new joint loss for the collection of all unobserved labels of an instance that uses label cardinality – either an estimate or known value – to encourage predictions consistent with the desired cardinality.

3. A bipartite matching algorithm designed for estimating the label cardinalities of all instances from current model outputs when an estimate of the cardinality distribution is available.

We empirically evaluate the value of the cardinality side information using standard SPMLL benchmark tasks. Already the first component that only requires crude estimates for mean and maximum cardinality achieves mean average precision (MAP) comparable to several recent SPMLL methods (Cole et al., 2021; Zhou et al., 2022; Kim et al., 2023; Chen et al., 2024). With richer cardinality information, we can obviously improve the results, and our focus is on quantifying the possible gain and studying the relationship between the accuracy of the cardinality estimates and the final classification performance. For example, for the commonly used NUS-WIDE data (Chua et al., 2009), knowing the cardinality instance is sufficient for almost matching the accuracy of a model trained on fully labeled data, and we get close even when only

assuming the cardinality distribution. Finally, we show that improving the accuracy of cardinality estimates directly translates to improved final model accuracy.

## 2 Related Work

To position the work within the broader SPMLL literature, we briefly outline the main approaches and some recent works. The most closely related works are on avoiding false negatives arising from the unobserved positive labels, resolved in our work by leveraging the cardinality information. As will be detailed later, the crudest approach, that assumes all unobserved labels to be negative, pushes the predictions to zero and results in severe problems with false negatives (Cole et al., 2021). A possible fix of maximizing entropy of the predictions (Zhou et al., 2022) overcompensates the issue, pushing predictions towards 0.5 that is considerably too high for most multi-label learning (MLL) tasks, resulting in false positives. Closest to our work is a method that regularizes the predictions using the expected label cardinality (Cole et al., 2021) while also using label smoothing; their regularizer can be seen as a special case of our general setup where different kinds of additional information on label cardinality are considered.

Also other approaches for avoiding false negatives have been studied, e.g. by preventing memorization of false negatives (Arpit et al., 2017; Kim et al., 2022) or by improving retrieval of positive labels by pseudo-labeling as commonly done in other weakly supervised settings (Chen et al., 2023; Wang et al., 2022). In SPMLL, Xie et al. (2022) considered pseudo-labeling combined with consistency regularization (Zhou et al., 2003; Laine & Aila, 2017) and Liu et al. (2023) employed a mutual information bottleneck for preserving label-specific characteristics during pseudo-labeling.

Other works have studied the related question of label imbalance; the SPMLL setup makes the common imbalance in MLL tasks more extreme. Several methods have been proposed for addressing class imbalance (Ridnik et al., 2021; Lin et al., 2020) and missing labels (Zhang et al., 2021) for standard MLL, with extensions for SPMLL. For instance, Chen et al. (2024) developed a noise-robust loss to account for the imbalance in SPMLL. We do not explicitly consider label scarcity, but cardinality information helps balancing the ratio of positive labels closer to the balance of the original MLL task. Finally, some SPMLL methods leverage domain-specific elements, such as class activation maps for images (Kim et al., 2023; Verelst et al., 2023). In contrast, our study focuses on the general task and does not consider any domain knowledge.

## 3 Problem Setup

**Multi-label learning** In MLL, each $q$-dimensional data point $\boldsymbol{x} \in \mathcal{X} = \mathbb{R}^q$ is associated with a binary label vector $\boldsymbol{y} \in \mathcal{Y} = \{0,1\}^C$ where $C$ is the number of classes. A fully labeled dataset $\mathcal{D} = \{(\boldsymbol{x}_i, \boldsymbol{y}_i) \mid 1 \le i \le N\}$ is available for training. Here, $y_i^j$ denotes the label of the $i$-th sample for the $j$-th class. $y_i^j = 1$ indicates that class $j$ is relevant (positive) for sample $i$, while $y_i^j = 0$ indicates irrelevance (negative). The objective is to learn a classifier $h : \mathcal{X} \mapsto [0,1]^C$ that minimizes a classification error, so that the outputs $\boldsymbol{f}_i = h(\boldsymbol{x}_i)$ are interpreted as probability of each class. Our work is agnostic to the functional form of $h$, and in the empirical experimentation we will always use the same model, ResNet-50 (He et al., 2016).

**Single-positive multi-label learning** We follow the common SPMLL formulation, matching e.g. (Xu et al., 2022). Instead of the full label vector $\boldsymbol{y}_i$ for data point $\boldsymbol{x}_i$, we observe a single-positive label vector $\boldsymbol{l}_i \in \{0,1\}^C$. Here, $l_i^j = 1$ indicates that the $j$-th class is the only observed positive label for $\boldsymbol{x}_i$, while $l_i^j = 0$ for all other classes, signifying that their true label values are unknown (not necessarily negative), i.e. $\sum_{j=1}^C l_i^j = 1$. We denote the set of unobserved classes for sample $i$ with $\mathbb{U}_i$. This results in a partially labeled dataset $\widetilde{D} = \{(\boldsymbol{x}_i, \boldsymbol{l}_i) \mid 1 \le i \le N\}$. Similar to MLL, the task here is to learn a classifier $h : \mathcal{X} \mapsto [0,1]^C$ for the original MLL task, predicting the full labels $\boldsymbol{y}_i$.

Our methods generalize for other partial labeling setups. We write the equations using a more general notation, where $k_i^o := \sum_{j=1}^C l_i^j$ labels are assumed observed, even though we run all experiments with $k_i^o = 1$.

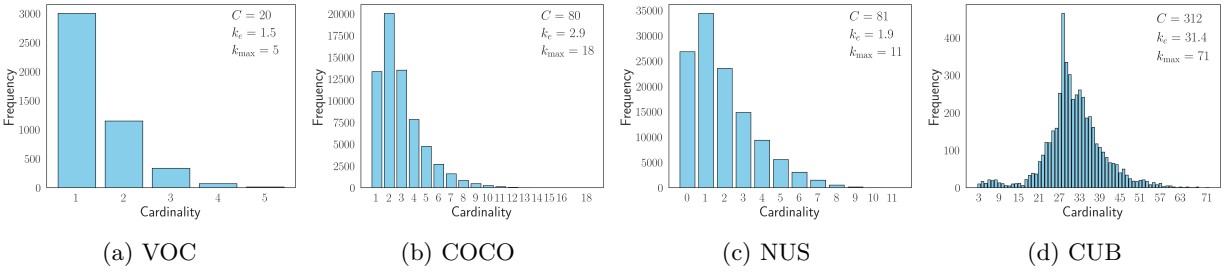

Figure 2: Cardinality distributions and statistics for MLL image benchmark datasets. Low-cardinality instances dominate when the labels are objects, but for the attribute-labels of CUB the distribution is clearly different.

**Label cardinality** This paper analyzes various ways of using label cardinality within the SPMLL framework. We denote the true label cardinality of instance $i$ as $k_i = \sum_{j=1}^{C} y_i^j$, and define three distinct categories of label cardinality information that can be available during training:

1. **Instance cardinality (IC)**: The most informative scenario, with known label cardinality $k_i$ for each instance $i$. We use it mostly as a theoretical construct to estimate the maximum information provided by cardinality information, but we also show how it can be estimated from the cruder statistics while learning the classifier.

2. **Cardinality distribution (CD)**: A more relaxed assumption is that we know (or can estimate) the discrete distribution $P(k)$ of the label cardinality. Figure 2 shows the CD for four common benchmark datasets, demonstrating a common pattern of decaying probability for higher cardinalities.

3. **Cardinality statistics (CS)**: If obtaining the full CD is not feasible, we can also use simple summary statistics of that, most commonly the maximum and mean label cardinality, $k_{\max} = \max_i k_i$ and $k_e = \frac{1}{N} \sum_i k_i$. Robust estimators, such as high quantiles instead of the maximum, can be used for additional robustness.

These categories represent a hierarchy of information content: CD implies CS, and both CD and CS can be computed from IC. In Section 5.2, we study how sensitive the proposed methods are for noisy estimates of cardinality information.

## 4 Method

We first briefly outline the standard SPMLL approach of factorizing the learning objective into separate losses for the observed and unobserved labels, and the issues of previously proposed losses for the unobserved labels. We then explain our technical solution for leveraging the cardinality information. We introduce two alternative losses for the unobserved labels $\mathbb{U}_i$, as the direct replacements for the previous proposals: (i) Section 4.2 provides a label-specific loss for cases where we only know the cardinality statistics, and (ii) Section 4.3 introduces a joint loss over the whole $\mathbb{U}_i$ that uses instance cardinalities as the learning target. In Section 4.4, we explain how instance cardinalities can be estimated based on the cardinality distribution and model predictions, and finally Section 4.5 explains a concrete SPMLL method using these components.

### 4.1 Background

For MLL with full label vector $\boldsymbol{y}_i$, the standard binary cross-entropy (BCE) loss is the most common choice. In the SPMLL setup, the typical losses (Chen et al., 2024; Cole et al., 2021) take the same factorized form

$$\mathcal{L}_{\text{SPMLL}} = \frac{1}{NC} \sum_{i=1}^{N} \sum_{j=1}^{C} \left( \mathbb{1}_{[l_i^j=1]} \mathcal{L}^+(f_i^j) + \mathbb{1}_{[l_i^j=0]} \mathcal{L}^u(f_i^j) \right), \tag{1}$$

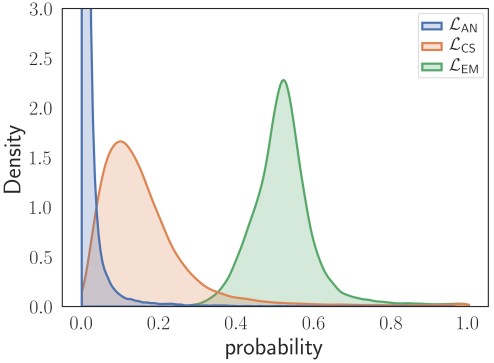 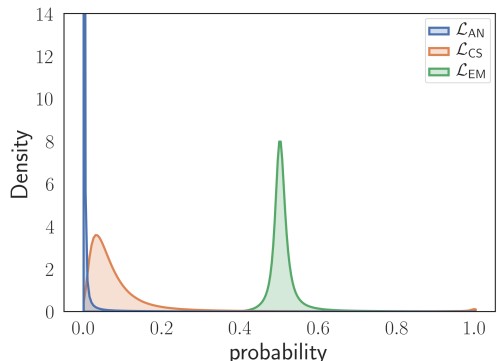

Figure 3: The distribution of model outputs $f_i^j$ for the unobserved labels on the VOC (left) and NUS (right) datasets. The $\mathcal{L}_{\text{AN}}$ loss causes false negatives by pulling predictions towards zero and $\mathcal{L}_{\text{EM}}$ causes false positives by pulling them towards 0.5. Our $\mathcal{L}_{\text{CS}}$ loss encourages predictions consistent with the cardinality statistics. For visual clarity, the y-axis is truncated at roughly 20% of the maximum density of $\mathcal{L}_{\text{AN}}$.

but use separate loss components, $\mathcal{L}^+(f_i^j)$ for observed positive labels and $\mathcal{L}^u(f_i^j)$ for the unknown ones. Here $\mathbb{1}$ is the indicator function selecting the correct loss for each predicted probability $f_i^j$.

Since $\boldsymbol{y}_i$ is unobserved, a naive choice is to replace it with the observed label vector. Substituting $\boldsymbol{l}_i$ for $\boldsymbol{y}_i$ in the BCE loss gives the Assumed Negative (AN) loss (Cole et al., 2021):

$$\mathcal{L}_{\text{AN}} = -\frac{1}{C} \sum_{j=1}^{C} \left( \mathbb{1}_{[l_i^j=1]} \log(f_i^j) + \mathbb{1}_{[l_i^j=0]} \log(1 - f_i^j) \right). \tag{2}$$

However, this loss tends to assign near-zero probabilities to all unknown labels ($l_i^j = 0$), as shown empirically in Figure 3. This both misleads the model severely and limits the information content of the predictions. To improve on this, Zhou et al. (2022) used entropy as $\mathcal{L}^u$ (denoting the total loss as $\mathcal{L}_{\text{EM}}$), but since the labels are binary variables, it effectively just changes the target for the probabilities, making them converge towards 0.5 instead of 0; see Figure 3. This implicitly suggests that we have prior probability around 0.5 for each individual class to be positive, which is considerably too high for typical MLL datasets with $k_i \ll C$ for all instances.

## 4.2 Unobserved Loss Based on Cardinality Statistics

Under this setting, we assume access to the maximum and mean label cardinality, respectively $k_{\max}$ and $k_e$. In practice, estimates of these quantities are sufficient, as demonstrated in Section B.1. To prevent the misinformation induced by the poor targets for $f_i^j$, we propose using $\mathcal{L}^u(f_i^j)$ satisfying three criteria:

1. Both very low and high $f_i^j$ incur a penalty – the model should not *a priori* be certain about the predictions for the unobserved labels.

2. Between these extremes, within a range specified soon, the loss should not favor any specific output – there is no information about which $f_i^j$ to prefer.

3. The range aligns with the statistical information about the frequency of the positive labels, as a way of encoding prior information about $f_i^j$.

Any loss satisfying these prevents the predictions from collapsing towards a pre-defined value and drives them to be on the right scale. This is as such useful – we will show in Section 5.2 that this loss achieves highly competitive performance – but informative $f_i^j$ will also be needed for our cardinality estimator described in Section 4.4.

Concretely, we use a quadratic penalty for $f_i^j$ outside the given lower and upper thresholds, $T_l$ and $T_u$. These thresholds are both determined based on the cardinality statistics to provide a useful prior for $f_i^j$. The upper threshold $T_u$ is set by distributing the maximum possible number of labels $k_{\max}$ (a known cardinality statistic) evenly across all unobserved classes, so that it is $(k_{\max} - 1)/(C - 1)$, where one is subtracted because of the already observed label. Outputs below the threshold do not provide evidence on the presence of a particular label, since they are consistent with a model that predicts each class equally likely for a sample of the highest cardinality. Any $f_i^j$ above the threshold means committing to that particular label and is penalized for.

We could, in principle, set the lower threshold $T_l$ similarly based on the minimum cardinality. However, this would make the threshold zero for most MLL datasets – even a single instance with only one label is enough to ensure this. It is essential to prevent zero probabilities that indicate strong commitment to non-presence of a particular class, and hence $T_l$ needs to be strictly non-negative. Any small constant is likely to work, and as a practical heuristic, we use $k_e$, the expected number of classes. As shown in Figure 2, it is often (but not always) small.

Building on this intuition, we formally define a new loss

$$\mathcal{L}_{\mathrm{CS}}^u(f_i^j) = \begin{cases} \lambda \left( f_i^j - \dfrac{k_{\max} - k_i^o}{C_i^u} \right)^2 & f_i^j > \dfrac{k_{\max} - k_i^o}{C_i^u} \\[2ex] \lambda \left( f_i^j - \dfrac{k_e - k_i^o}{C_i^u} \right)^2 & f_i^j < \dfrac{k_e - k_i^o}{C_i^u} \\[2ex] 0 & \text{Otherwise} \end{cases} \tag{3}$$

where $k_e$ and $k_{\max}$ are the average and maximum number of positive labels per instance, $k_i^o$ is the number of observed positive labels for instance $i$ (which is $\forall_i k_i^o = 1$ in standard SPMLL), $C_i^u = C - k_i^o$ is the number of unobserved classes for that instance, and $\lambda$ is a scaling hyperparameter. Outside the non-penalized range defined by the thresholds, we apply a quadratic penalty commonly used in loss function design, but note that other monotonically increasing penalty functions could be used as well. The general loss defined in Equation 1 using $\mathcal{L}_{\mathrm{CS}}^u$ is denoted by $\mathcal{L}_{\mathrm{CS}}$.

### 4.3 Unobserved Loss Based on Instance Cardinality

The IC hypothesis assumes that we know $k_i$, the number of positive labels for each instance $i$. This implies $k_i^u = k_i - k_i^o$ labels within the set of unobserved classes $\mathbb{U}_i$, again with $k_i^o = 1$ for the SPMLL setup. Next, we explain how this information can be used for constructing a joint loss over all of the unobserved labels for a given instance. Instead of factorizing the loss over all of the labels, as in the case of all previously proposed $\mathcal{L}^u$, we encourage the joint predictions for the unobserved labels to be consistent with the instance cardinality. The overall loss is

$$\mathcal{L}_{\mathrm{IC}} = \frac{1}{NC} \sum_{i=1}^{N} \sum_{j=1}^{C} \mathbb{1}_{[l_i^j = 1]} \mathcal{L}^+(f_i^j) + \frac{\alpha}{N} \sum_{i=1}^{N} \mathcal{L}_{\mathrm{IC}}^u(\boldsymbol{f}_i), \tag{4}$$

where $\alpha$ is a scaling hyperparameter and $\mathcal{L}_{\mathrm{IC}}^u$ is the new loss defined jointly for the set $\mathbb{U}_i$ of the unobserved labels.

The $\mathcal{L}_{\mathrm{IC}}^u$ is designed to achieve two goals. First, it encourages the total probability mass assigned for probable classes (the classes for which the current $f_i^j$ are high, to be defined formally soon) to align with the desired cardinality $k_i^u$. Second, it drives the probabilities for the remaining classes towards zero. Intuitively, this both ensures the model retains sufficiently high probability for enough classes while allowing it to commit to some classes being negative. Formally,

$$\mathcal{L}_{\mathrm{IC}}^u(\boldsymbol{f}_i) = \left( \left( \sum_{j \in \Omega_i} f_i^j \right) - k_i^u \right)^2 + \beta \left( \sum_{j \notin \Omega_i} f_i^j \right)^2. \tag{5}$$

Here $\beta$ is a scaling hyperparameter, $\Omega_i \subset \mathbb{U}_i$ is the set of indices corresponding to the $M_i = |\Omega_i|$ classes with the highest predicted probabilities for instance $i$, and $k_i^u$ is the corresponding target cardinality – the number of unobserved classes that should be positive. The first term encourages the sum of the top $M_i$ predicted probabilities to stay close to $k_i^u$, whereas the second term pushes the remaining labels towards low probability. Note that, if $k_i^u = 0$, we have $\Omega_i = \emptyset$ and only the second term remains. That is, we explicitly consider all unobserved labels as negative, achieving the intuitive property mentioned in Section 1.

**Remark: Selecting $M_i$** The general formulation leaves open the question of how many of the largest entries are included in the summation. We should always have $M_i \geq k_i$ to ensure we collect the probability mass over at least $k_i$ classes. In practice, we want $M_i > k_i$ to prevent false negatives when the current model predictions are inaccurate or $k_i$ is not necessarily exact. This is because all $f_i^j \notin \Omega_i$ are pushed towards zero. We use two alternative choices in our experiments, to avoid enforcing such false negatives: $M_i = 2k_i$ when we know $k_i$ reliably, and $M_i = k_{\max} - k_i^o$ when we need to estimate the cardinalities.

**Remark: Scaling probabilities** Equation 5 aligns the sum of the model outputs with $k_i$, but any monotonic transformation of $f_i^j$ could be used instead. Since we typically have $f_i^j < 1$ even for the correctly predicted classes, it may be beneficial to use a transformation that saturates to one faster. We use a scaled sigmoid that is steeper than the one used for actual model outputs. If we denote by $\boldsymbol{z}_i$ the logits that are transformed for $\boldsymbol{f}_i$, we use in place of $f_i^j$ transformed outputs $\tilde{f}_i^j = \frac{1}{1+\exp\{-a(\boldsymbol{z}_i)\}}$ where $a > 1$.

## 4.4 Estimating Instance Cardinalities

When the label cardinality $k_i$ for each instance is not known, we cannot directly use the loss in Equation 5. However, we can replace the known $k_i$ with any estimate $\hat{k}_i$ instead. In principle, any estimator could be used, for instance a model that takes as input either the input vectors $\boldsymbol{x}_i$ or the model predictions $\boldsymbol{f}_i$ and is trained in supervised fashion to provide the estimate. However, this would require knowledge of the true cardinalities $k_i$ for some training instances and a dedicated training process. Next, we introduce a novel algorithm for estimating the instance cardinalities, requiring only knowing the cardinality distribution, a probability vector of $P(k)$. The method is computationally light and does not require any training or other supervision; in particular, we do not need to know the true $k_i$ for any instance.

We formulate the problem as perfect bipartite matching between two sets of $N$ elements (Cormen et al., 2022). The first set corresponds to the instances, represented by a vector $\boldsymbol{s} \in \mathbb{R}^N$ of scores. For each instance, we define $s_i = \sum_{j \in \Omega_i} f_i^j$ with $M_i = k_{\max}$, corresponding to the quantity that is pushed towards the cardinality in Equation 5. The idea is to now use the score as the basis for estimating the cardinality itself. Note that the score is defined based on the current model predictions $\boldsymbol{f}_i$, meaning we assume they carry information about the cardinality. This will be the case for models trained using the proposed $\mathcal{L}_{\mathrm{CS}}$ and $\mathcal{L}_{\mathrm{IC}}$ losses, but not e.g. for $\mathcal{L}_{\mathrm{AN}}$.

The other set is constructed based on the cardinality distribution $P(k)$, so that it has the right frequency for each cardinality in increasing order. We denote by $\boldsymbol{v} \in [0, ..., k_{\max}]^N$ a vector where the first $P(k=0)N$ entries are set to 0, the next $P(k=1)N$ entries to 1, etc. It has the right cardinality distribution by construction.

We find the perfect bipartite match $\Pi$, a permutation over the $N$ entries of the latter set, that minimizes $\sum_i (s_i - \Pi_i \boldsymbol{v})^2$. That is, we assign for each instance a single entry in $\boldsymbol{v}$ and hence a cardinality estimate $\hat{k}_i = \Pi_i \boldsymbol{v}$ so that the total distance between $s_i$ and $\hat{k}_i$ is minimized. Even though bipartite matching has cubic complexity for the general case (Kuhn, 1955), our problem is a simplified special case of Euclidean bipartite matching (Agarwal & Varadarajan, 2004; Arora, 1997) with one-dimensional points – each element is represented by a non-negative scalar ($s_i$ or $v_i$). For this problem, we can find the global optimum in $\mathcal{O}(N \log N)$ time with an algorithm that sequentially matches the candidates (see Appendix A for a proof): Iteratively assign the smallest cardinality for the sample with the smallest $s_i$ and remove the corresponding elements from the sets. Since $\boldsymbol{v}$ is ordered by construction, this can be done by sorting $s_i$ and collecting the sorting order in $\Pi$.

### 4.5 Whole Algorithm

Having introduced the technical elements, we now describe a practical SPMLL algorithm. We note, however, that each of the elements could be used also in other ways and this specific algorithm is introduced as an example for the purpose of the comparisons. Both of the unobserved label losses (Equation 3 and 5) can be used as plug-in replacements in other SPMLL methods, and the cardinality estimation algorithm could be replaced with other estimators.

For the **CD setup**, where the distribution of cardinalities is assumed known, our general algorithm works as follows and is denoted by `CS ⇒ CD` in the results:

1. First optimize $\mathcal{L}_{\mathrm{CS}}$ for some epochs, for initialization of the model so that $\boldsymbol{f}_i$ start being informative, both about the class predictions and about the cardinalities.

2. Solve $\hat{k}_i$ with the sorting algorithm described in Section 4.4, using the known CD and $\boldsymbol{s}$, computed based on the current $\boldsymbol{f}_i$ for all instances, as the input.

3. Optimize $\mathcal{L}_{\mathrm{IC}}$ for more epochs using $\hat{k}_i^u = \hat{k}_i - k_i^o$ as the target in Equation 5. After each epoch, re-estimate $\hat{k}_i$ using the current predictions as in step 2.

For the **CS setup**, the algorithm simplifies to only the first step above and is denoted as `CS` in the results. For the **IC setup**, we use the known $k_i$, skipping the estimation of $\hat{k}_i$, and denote it by `CS ⇒ IC`.

## 5 Experiments & Results

### 5.1 Experimental Setup

**Data** We use four common SPMLL benchmark datasets for evaluation: Pascal VOC 2012 (VOC) (Everingham et al., 2015), MS COCO 2014 (COCO) (Lin et al., 2014), NUS-Wide (NUS) (Chua et al., 2009), and CUB-200-2011 (CUB) (Wah et al., 2011). The datasets are used as in the previous literature, e.g. using the 312 binary attributes of CUB as target labels as in Cole et al. (2021); Zhou et al. (2022), with full details disclosed in Appendix C.2. The cardinality distributions and statistics are shown in Figure 2.

The test setup follows closely previous SPMLL works, e.g. Cole et al. (2021); Zhou et al. (2022). We randomly split each original training data into our training and validation sets, using 20% for validation. The original validation data is used as the test set. The SPMLL training data is formed by choosing the observed label uniformly at random from the set of true positive labels for each instance, assumed also by the comparison methods. That is, we do not consider the more general setups with e.g. class-specific weighting studied in the broader positive and unlabeled literature (Elkan & Noto, 2008). The validation and test sets are assumed fully labeled, as in previous works, to enable a fair comparison against the baselines and reduce random variation; see Section 6 for discussion.

**Comparison methods and implementation details** We compare against several recent SPMLL methods: ROLE (Cole et al., 2021), EM+APL (Zhou et al., 2022), LL-R + BoostLU (Kim et al., 2023), and GR Loss (Chen et al., 2024). In addition, we report the results for the weak baseline of AN loss (Cole et al., 2021) and the upper bound of training the same model on fully labeled data.

We use the same classifier for all methods, with ResNet-50 (He et al., 2016) backbone pretrained on ImageNet (Russakovsky et al., 2015) and technical details matching the implementation of Cole et al. (2021); the details of image preprocessing, data augmentation, and the architecture are provided in Appendix C.1. The models are trained end-to-end using the specific loss of each method with Adam (Kingma & Ba, 2015) and the performance is evaluated using MAP averaged over the $C$ classes. For all the methods, the hyperparameters are determined with a grid search using validation MAP as the metric, as detailed in Appendix C.4. For simplicity, we use fixed batch sizes from Zhou et al. (2022) and learning rate from Chen et al. (2024) for all the methods, focusing on validating the method-specific parameters.

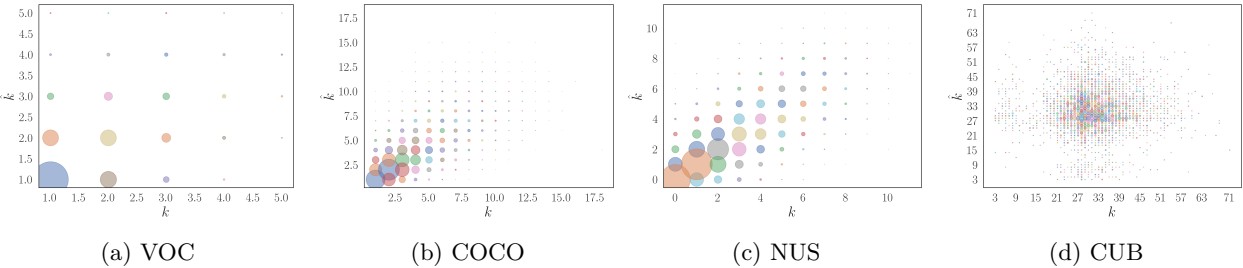

(a) VOC        (b) COCO        (c) NUS        (d) CUB

Figure 4: Cross-plot of estimated vs true label cardinality, with the area of the circles indicating the count; the colors are for visual separation and have no meaning. Figure 10 in Appendix shows an alternative representation of this data using the conditional distribution of the predicted cardinalities.

## 5.2 Experiments

We conduct three main experiments. The first contrasts the proposed methods against the baselines, the second quantifies the value of the cardinality information, and the third validates the methods are robust with respect to estimation error when CS or CD needs to be estimated from data.

**Experiment 1** Table 1 reports the test set MAP for three variants of our method and all comparison methods. The results are averaged over three runs. The key comparison is between CS and the previous SPMLL methods. Except for the COCO data, where LL-R + BoostLU (Kim et al., 2023) stands out, it is comparable or better than all other methods. The method is easy to implement, only requiring replacing the loss for the unobserved labels with one that encourages $f_i^j$ to remain on a reasonable scale (see Figure 3), and only very crude cardinality information is needed to set the two thresholds in Equation 3; we will show the method is insensitive to a specific choice in Section B.1.

The other two variants of our method should not be directly compared with the rest, since they require more information. Instead, the attention should be placed on the difference between the three proposed methods. For COCO and NUS, we observe the expected effect: Having access to CD shows as clear improvement, and knowing IC helps more as it should. For NUS, it is worth noting that CS $\Rightarrow$ IC effectively reaches even the accuracy of the fully labeled case. For VOC data all SPMLL methods reach essentially the accuracy of the full label upper bound, and hence there is no room for improvement from the use of cardinality information. For CUB, in turn, we observe that even knowing the IC does not help, which already implies that the CD cannot help either.

Figure 4 illustrates the cardinality estimators. The area of each circle represents the frequency of a $(k, \hat{k})$ pair, and a perfect estimator would result in a diagonal line where the sizes follow the cardinality distribution. The estimates for COCO and NUS are good, consistent with the improvement in MAP in Table 1. For CUB, we also confirm the numerical result; as knowing $k$ does not even help learning a better model, it is natural that we cannot estimate them either based on the model outputs.

**Experiment 2** Exact label cardinalities are naturally more informative than the ones estimated based on the cardinality distribution. To better understand the difference in information content, we study two related questions: (Q1) How improvements in cardinality estimates translate to improvements in MAP, and (Q2) How accurate the estimator needs to be to still be useful.

We study both questions using a simulation experiment. We run the CS $\Rightarrow$ CD method, but instead of using the scores $s_i = \sum_{j \in \Omega_i} f_i^j$ as the inputs for the estimator described in Section 4.4, we now use $s_i = k_i + \epsilon$, where $\epsilon \sim \mathcal{N}(0, \sigma^2)$. That is, we do not use the actual model outputs for estimating the cardinalities but assume access to a noisy oracle that allows generating estimates of varying quality in a controllable fashion: if $\sigma^2 = 0$ then we recover the IC scenario of perfect estimates, whereas with $\sigma_2 \to \infty$ we converge to a random permutation of the cardinalities. Note that the cardinality distribution constraint is still satisfied even in this case, by the virtue of the estimation algorithm.

Table 1: Test MAP for all methods. The `CS` method can be directly compared against the previous methods, but our `CS ⇒ CD` and `CS ⇒ IC` methods assume more information.

| Method | VOC | COCO | NUS | CUB |
|---|---|---|---|---|
| Full label | $89.56 \pm 0.35$ | $76.43 \pm 0.09$ | $52.12 \pm 0.09$ | $32.73 \pm 0.20$ |
| AN | $85.51 \pm 0.41$ | $64.33 \pm 0.10$ | $42.68 \pm 0.09$ | $18.65 \pm 0.31$ |
| ROLE | $88.10 \pm 0.16$ | $66.89 \pm 0.16$ | $41.74 \pm 0.21$ | $14.52 \pm 0.47$ |
| EM+APL | $89.19 \pm 0.29$ | $70.92 \pm 0.23$ | $47.56 \pm 0.20$ | $21.13 \pm 0.60$ |
| LL-R+BoostLU | $88.81 \pm 0.24$ | $73.11 \pm 0.21$ | $49.76 \pm 0.51$ | $19.75 \pm 0.23$ |
| GR Loss | $89.38 \pm 0.12$ | $71.33 \pm 0.21$ | $47.08 \pm 0.43$ | $20.69 \pm 0.28$ |
| CS (ours) | $88.97 \pm 0.03$ | $71.35 \pm 0.10$ | $49.23 \pm 0.45$ | $21.56 \pm 0.08$ |
| CS ⇒ CD (ours) | $89.10 \pm 0.17$ | $72.07 \pm 0.08$ | $50.26 \pm 0.08$ | $21.58 \pm 0.21$ |
| CS ⇒ IC (ours) | $89.69 \pm 0.16$ | $73.45 \pm 0.12$ | $50.92 \pm 0.48$ | $21.73 \pm 0.35$ |

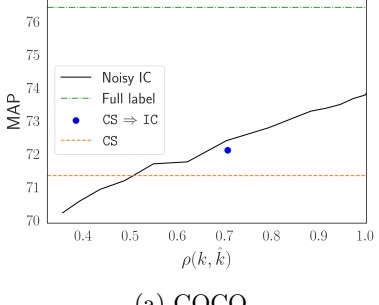
(a) COCO

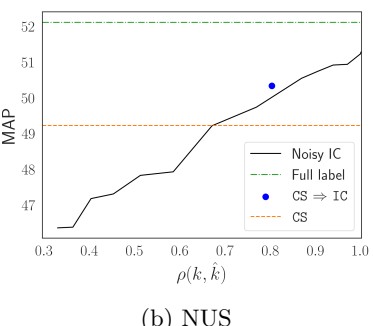
(b) NUS

Figure 5: The MAP score improves approximately linearly when the cardinality estimates are improved (black), and the real estimator (blue point) aligns well with the simulation model. MAP for full label (green) and CS (orange) setups are provided as scale reference.

We repeat this for a range of $\sigma^2$, from very poor estimates to perfect cardinalities. We quantify the quality by computing the Pearson correlation between $\hat{k}$ and $k$ over the $N$ samples, and evaluate the MAP as well. Figure 5 shows that the relationship of these two is essentially linear for both COCO and NUS, the two data sets with most potential for leveraging cardinality information. That is, we confirm that improving the estimator results in improved MAP, and additionally learn that the transition is gradual – all improvements in cardinality estimates are directly reflected in the MAP score, answering Q1. The answer to Q2 is given by the point where this trend intersects the accuracy obtained with `CS`: For COCO, any estimator with $\rho(\hat{k}, k) > 0.5$ helps, and we need $\rho(\hat{k}, k) > 0.65$ for NUS.

We can also compute $\rho(\hat{k}, k)$ for the real method where $s_i = \sum_{j \in \Omega_i} f_i^j$ is used for estimating $\hat{k}_i$, corresponding to the results visualized in Figure 4. For both COCO and NUS, the result, indicated by the pair of the correlation and MAP, is close to the trend line for the simulation. This confirms the simulation process mimics the real estimation error and is hence informative for real use cases.

**Experiment 3** For the experiments above, we used exact CS and CD to isolate the effect of possible estimation error, but in real use-cases, they need to be estimated. We next show that the method is extremely robust with respect to the estimation errors. The `CS` method only requires $\hat{k}_{\max}$ for setting $T_u$, and already an extremely crude guess is enough. For any $\hat{k}_{\max}$ between 0.5 and 2.5 times the real $k_{\max}$, we observed virtually no change in MAP. For instance, for COCO, the MAP scores remain between 70.76 and 71.35; see Figure 7 in Appendix for full details.

CD can be estimated in multiple ways. Figure 6 shows how MAP develops when estimating it from a sample of $S$ instances for which $k_i$ is annotated (note that we still do not require fully annotated samples), using the crudest possible estimator: Set $\hat{k}_{\max}$ to a maximum of the largest $k$ in the sample and a user-defined upper bound (we used $1.2k_{\max}$ as an example) and estimate $P(k)$ as the histogram of the $S$ samples with additive

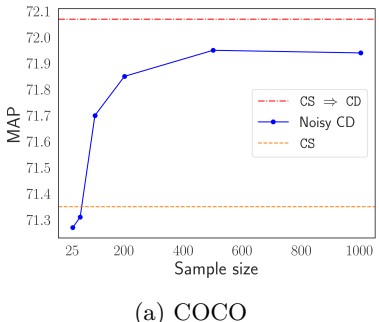

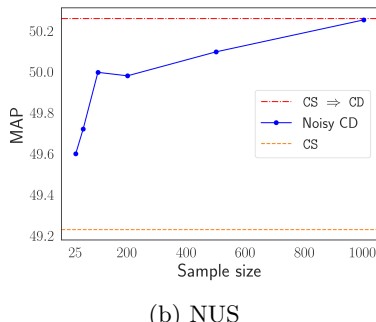

(a) COCO                                    (b) NUS

Figure 6: MAP progressively improves when using more accurate estimates of cardinality density, here shown as a function of sample size $S$ with known instance cardinalities.

Table 2: Test MAP for all methods using validation data that contains only single-positive label per instance, referred to as "Single" in the table. For ease of comparison, the results on fully annotated validation data ("Full") from Table 1 are also included.

| Method | Full | | Single | |
|---|---|---|---|---|
| | COCO | NUS | COCO | NUS |
| AN | $64.33 \pm 0.10$ | $42.68 \pm 0.09$ | $64.13 \pm 0.21$ | $42.33 \pm 0.54$ |
| ROLE | $66.89 \pm 0.16$ | $41.74 \pm 0.21$ | $66.95 \pm 0.09$ | $41.16 \pm 0.69$ |
| EM+APL | $70.92 \pm 0.23$ | $47.56 \pm 0.20$ | $70.76 \pm 0.18$ | $47.56 \pm 0.20$ |
| LL-R+BoostLU | $73.11 \pm 0.21$ | $49.76 \pm 0.51$ | $70.71 \pm 0.18$ | $48.31 \pm 0.91$ |
| GR Loss | $71.33 \pm 0.21$ | $47.08 \pm 0.43$ | $71.06 \pm 0.13$ | $47.38 \pm 0.02$ |
| CS (ours) | $71.35 \pm 0.10$ | $49.23 \pm 0.45$ | $71.19 \pm 0.19$ | $49.02 \pm 0.19$ |

Laplace smoothing. For NUS, already $S = 25$ is enough for clearly improving over CS and for both data sets increasing $S$ improves MAP as it should. See Appendix B.2 for further details and analysis, showing how MAP relates to estimation error and thus providing information on how alternative estimators (e.g. elicitation from a domain expert) would work.

## 6    Discussion

**Evaluation**   Some remarks are useful for correct interpretation of the results. All methods in Table 1 were run by us to ensure consistency, explaining differences to the scores reported in previous literature. The MAP scores are mostly in line with the ones in the original publications, but for the GR Loss Chen et al. (2024), we could not replicate the good performance, despite using their code. As explained in Section 5.1, we compared only against baselines that use the same unmodified ResNet-50 backbone, to ensure the differences are due to the loss and algorithm changes. Some works report higher MAP for these benchmark datasets, but the numerical accuracies cannot be directly compared since the architecture has significant effect on the overall performance. For instance, Xie et al. (2022) reports strong results but also notably higher MAP even for the AN and ROLE baselines, confirming part of the improvement is because of the different base model.

As in all comparison methods, we used fully labeled validation set for determining the hyperparameters to keep results comparable with the previous literature. This slightly exaggerates all accuracies due to the use of data that would not be available in real use cases, but the bias is similar for all methods and in general small. Note that CS, the method we compare against the previous ones, has fewer hyperparameters than most and cannot gain unfair advantage due to tuning. To validate this empirically, Table 2 reproduces Table 1 for the two interesting datasets using a validation set with only single-positive label per instance. The results are highly similar, except for LL-R+BoostLU that has several hyperparameters dropping notably in MAP. Our CS method now outperforms all others on both datasets.

**Value of cardinality information** The empirical results can be summarized as follows: (a) CS alone is a strong SPMLL method and in practice does not require any additional information; it uses the statistics for setting the thresholds but works even with 50% estimation error for the maximum cardinality. (b) Cardinality distribution, either known or estimated from a very small sample, improves MAP compared to CS. (c) Improved estimation of the instance cardinalities improves overall performance, with approximately linear dependency between the estimation quality (measured by correlation) and MAP. (d) The improvements in MAP are typically not large in absolute terms, but it is important to contrast the gain to what could be achieved: For VOC and NUS the results are already effectively at the level of the fully-labeled model.

Finally, we note that the CS ⇒ IC result gives an upper bound for MAP when optimizing the loss in Equation 5. However, it is not necessarily the best way to use this information; e.g. various pseudo-labeling methods (Liu et al., 2023) could be boosted with cardinality information.

**Generality** We showed consistent good performance on four datasets, but with clear differences in how much the cardinality information helps. For NUS and COCO, we can estimate cardinalities well and MAP improves clearly, but for CUB even known cardinalities do not help. We do not have a clear interpretation for this – it may relate to the shape of the cardinality distribution, or to the data itself. For instance, leveraging the cardinality information will be difficult if $k_i^u$ is almost conditionally independent of $y_i^j$ for $j \in \mathbb{U}_i$ given the observed labels. This is more likely when $k_i^u \gg k_i^o$, which holds for CUB but not for the other data sets. For VOC, there is no room for improvement; already some methods not using any cardinality information match the MAP of the model trained on full labels.

In this work, we focused on the SPMLL setup, but wrote the equations for more general partial labeling scenarios. Appendix D demonstrates that the method works unmodified also in more general scenarios, by showing results for an experiment where two positive labels are assumed known for each sample, when applicable. The accuracy of all methods naturally improves as it should, but the improvement from using cardinality information remains similar as in the SPMLL setup.

Our technical elements could be integrated into other SPMLL methods. First, $\mathcal{L}_{\mathrm{CS}}$ could be used as strong initialization for any method, not requiring any modifications for the methods themselves. Moreover, since the objective functions in many prior works are factorized as in Equation 1, i.e. the loss term for unlabeled classes $\mathcal{L}^u$ is an additive component to the entire loss function, $\mathcal{L}_{\mathrm{CS}}^u$ can be directly used as a plug-in replacement for their unsupervised components, again without modifying the rest of the method. Extensions where the cardinality information is combined with alternative means of improving SPMLL are particularly interesting. For example, EM+APL (Zhou et al., 2022) uses pseudo-labeling to improve the learning, and we could replace their entropy-based regularization with $\mathcal{L}_{\mathrm{CS}}^u$ in an attempt to guide the pseudo-labeling with cardinality estimates. Recently Hu et al. (2025) proposed using active learning to select partially labeled samples for full annotation, and cardinality information could potentially be used for guiding this choice as well. Evaluation of these ideas is a promising avenue for future work.

## 7 Conclusion

We introduced a novel problem formulation where varying degrees of side information on the label cardinality is used in the MLL tasks with only partial labeling, focusing on the special case of SPMLL. We characterized three concrete forms of side information, explained how they can be used for improving SPMLL performance, and analyzed the problem. Already a minor modification for the loss assumed for unobserved labels, requiring only crudely estimated statistics of the cardinality distribution, gives a well-performing SPMLL method. The accuracy can be progressively improved by leveraging either a distribution of the label cardinalities or instance cardinality estimates.

**Acknowledgments**

This work was supported by the Research Council of Finland project 353441 and the Flagship programme: Finnish Center for Artificial Intelligence, FCAI. The authors acknowledge CSC – IT Center for Science, Finland, for computational resources.

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

## A  Proof for Perfect Bipartite Matching with Greedy Algorithm

Section 4.4 described that the instance cardinalities are estimated as $\hat{k}_i = \Pi_i \boldsymbol{v}$, where $\Pi$ is a permutation of a cardinality vector $\boldsymbol{v}$ satisfying the cardinality distribution. Next, we show that the permutation $\Pi$ that minimizes the objective $\mathcal{B} = \sum_i (s_i - \Pi \boldsymbol{v})^2$ can be found by sorting the entries of $\boldsymbol{s}$. Let's start by re-writing the objective as

$$\mathcal{B} = (\boldsymbol{s} - \Pi \boldsymbol{v})^T (\boldsymbol{s} - \Pi \boldsymbol{v})$$
$$= \boldsymbol{s}^T \boldsymbol{s} + \boldsymbol{v}^T \Pi^T \Pi \boldsymbol{v} - 2 \boldsymbol{s}^T \Pi \boldsymbol{v}.$$

Since $\Pi^T \Pi = I$, the first two terms do not depend on $\Pi$. Consequently, the task is equivalent to maximizing $2\boldsymbol{s}^T \Pi \boldsymbol{v}$. According to the rearrangement inequality (Hardy et al., 1952), this is achieved when the elements are greedily paired according to their magnitude: the largest element of $\boldsymbol{s}$ is matched with the largest element of $\boldsymbol{v}$, and so on. Since $\boldsymbol{v}$ is in increasing order by construction, the solution is found by sorting $\boldsymbol{s}$ analogously, with obvious $\mathcal{O}(N \log N)$ complexity.

## B  Sensitivity Analysis

### B.1  Sensitivity to $k_{\max}$

The largest cardinality $k_{\max}$ is a dataset-dependent parameter that primarily determines the upper threshold $T_u$ in Equation 3 of the CS method. Figure 7 illustrates the CS method's insensitivity to variations in the maximum value $\hat{k}_{\max}$, ranging from 0.5 to approximately 2.5 times the actual $k_{\max}$, without significantly affecting performance. Alternatively, it is possible to use more robust statistics such as the 95% quantile estimator, with the overall robustness accounting for the small bias caused by the estimator.

In Equation 3, the lower threshold $T_l$ is similarly determined by $k_e$. In practice, the method works with any small number that is strictly non-zero, and setting it at $k_e$ that for most multi-label datasets is much smaller than the number of classes $C$ is simply a convenient practice.

### B.2  Sensitivity to CD Estimation Error

For Experiment 3, we estimate the CD from a set of $S$ samples for which $k_i$ is known. We denote by $m_k$ the count of a given cardinality $k$ in the sample and estimate the density as

$$\hat{P}(k) = \begin{cases} \dfrac{m_k + \delta}{\sum_{\nu=1}^{\hat{k}_{\max}} (m_\nu + \delta)} & k \le \hat{k}_{\max} \\ 0 & \text{otherwise,} \end{cases}$$

where $\delta$ is a smoothing factor and $\hat{k}_{\max}$ is an estimate of the maximum cardinality. For the results reported in Figure 6, we used $\delta = 0.1$ and $\hat{k}_{\max} = 1.2 k_{\max}$, but the results would be highly similar for a broad range of other choices. Note that when estimating $P(k)$ with an initial $\hat{k}_{\max}$ that is too small, there may be $k_i$ in the sample that exceed the presumed maximum. In such cases, $\hat{k}_{\max}$ should be re-set to the largest $k_i$ in the sample.

Figure 8 provides a complementary analysis. For each estimate, we compute the estimation error as Kullback-Leibler (KL) divergence between the estimate and the true cardinality, and here show the MAP as a function of the estimation error. This illustration provides information on how alternative estimators with similar estimation error would work.

### B.3  Sensitivity to $M_i$

The parameters $M_i$ determine how many of the largest probabilities $f_i^j$ are summed over in the joint loss Equation 5 and when estimating the cardinalities. Figure 9 shows the results are not sensitive to the choice, plotting the model performance as a function of $M_i$, and plotting the MAP score normalized with the highest

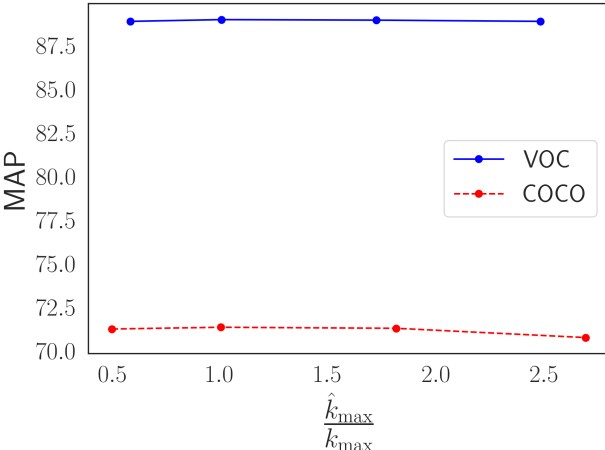

Figure 7: Sensitivity analysis of $k_{\max}$ and equivalently of the upper threshold $T_u$ in the CS method.

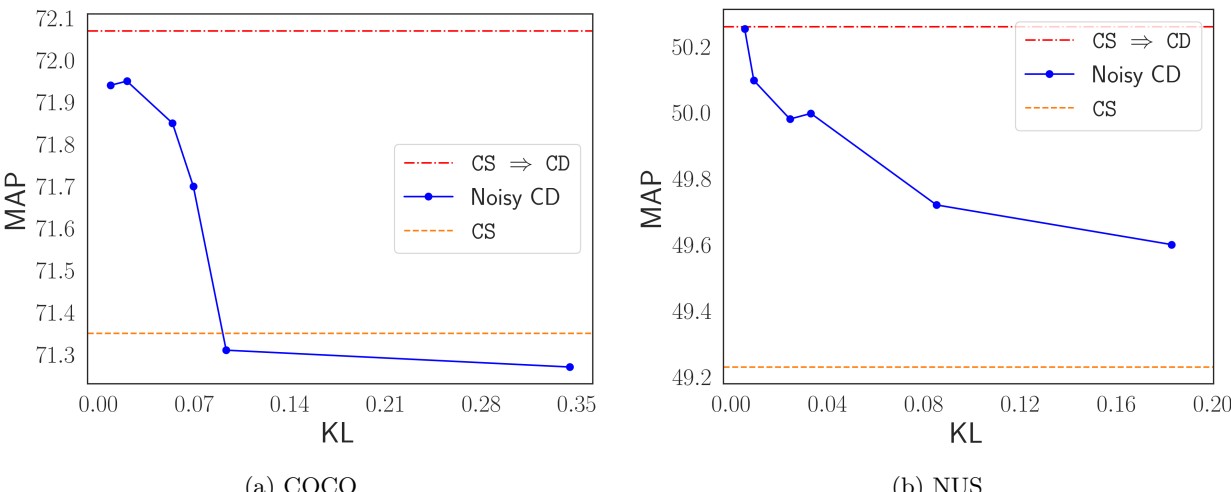

(a) COCO                  (b) NUS

Figure 8: Results of Figure 6 presented from an alternative perspective, as a function of the estimation error (KL divergence between the estimate and true cardinality density).

score. Except for $M_i = k_i$ that results in a minor reduction in MAP due to creating false negatives, the results are extremely consistent over the choice with at most 0.5% reduction in MAP. The choice of $M_i = k_{\max} - k^o$ that only depends on the maximum cardinality is effectively as good as any $M_i$ that depends on the instance cardinality $k_i$.

## C  Experimental Details

### C.1  Model Architecture

We use a ResNet-50 backbone pre-trained on ImageNet. Following the same image processing techniques as in all comparison methods, we resize all input images to $448 \times 448$ pixels and apply data augmentation of random horizontal flipping (probability 0.5) during training. The output layer of ResNet-50 is replaced by a global average pooling (Lin, 2013) followed by a fully connected layer, and the output dimension is adjusted to match the number of classes in the target dataset.

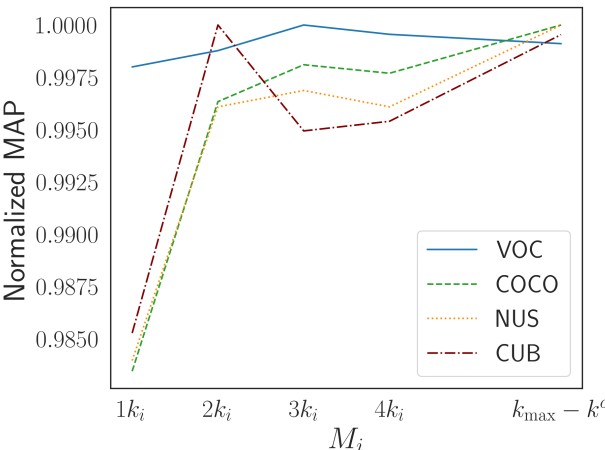

Figure 9: Sensitivity analysis of $M_i$ for the CS $\Rightarrow$ IC method. Normalized MAP scores are computed by dividing the MAP by the highest MAP for each dataset.

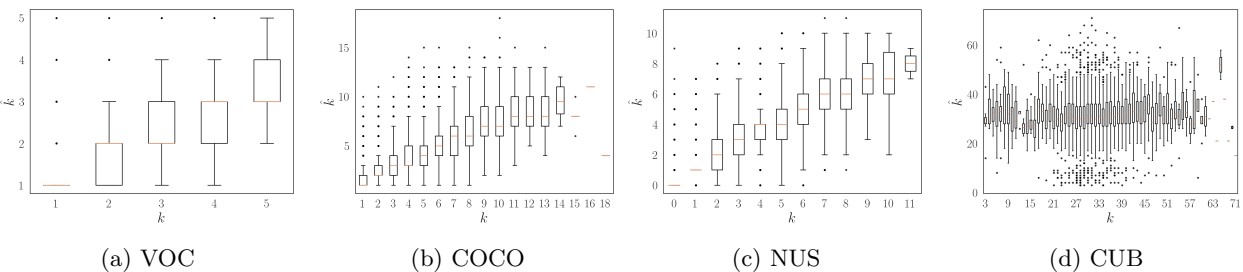

(a) VOC $\qquad\qquad$ (b) COCO $\qquad\qquad$ (c) NUS $\qquad\qquad$ (d) CUB

Figure 10: The conditional distribution of predicted cardinalities in Experiment 1.

## C.2 Datasets

Table 3 provides full details about the datasets, indicating both the standard information (number of images and $C$) and the cardinality statistics $k_e$ and $k_{\max}$.

Table 3: The details of benchmark datasets used in our experimentation.

| Statistics | | VOC | COCO | NUS | CUB |
|---|---|---|---|---|---|
| C | | 20 | 80 | 81 | 312 |
| # Images | Training | 4574 | 65665 | 120000 | 4795 |
| | Validation | 1143 | 16416 | 30000 | 1199 |
| | Test | 5823 | 40137 | 60260 | 5794 |
| $k_e$ | Training | 1.46 | 2.94 | 1.89 | 31.4 |
| | Validation | 1.46 | 2.92 | 1.93 | 31.52 |
| | Test | 1.43 | 2.91 | 1.88 | 31.53 |
| $k_{\max}$ | Training | 5 | 18 | 11 | 71 |
| | Validation | 5 | 16 | 11 | 69 |
| | Test | 5 | 15 | 13 | 72 |

## C.3 Training

For CS and CS $\Rightarrow$ CD, we train our model for 10 epochs and 20 epochs for CS $\Rightarrow$ IC, using Adam optimizer and established training parameters from prior work without further optimization. Following Zhou et al.

Table 4: Best hyperparameters for each method. The parameters $\alpha$ and $\beta$ used in Equation 4 and 5 are computed based on $\gamma$, $\eta$ and $\phi$ using Equation 6 and 7.

| Parameter | Method | Dataset | | | |
|---|---|---|---|---|---|
| | | VOC | COCO | NUS | CUB |
| $\lambda$ | CS | 0.35 | 0.15 | 0.1 | 0.01 |
| $e_w$ | CS $\Rightarrow$ CD | 6 | 6 | 5 | 5 |
| | CS $\Rightarrow$ IC | | | | |
| $a$ | all | 2 | 3 | 2 | 2 |
| $\gamma$ | CS $\Rightarrow$ CD | $1e^{-4}$ | $8e^{-5}$ | $4e^{-3}$ | $6e^{-6}$ |
| | CS $\Rightarrow$ IC | $9e^{-3}$ | $6e^{-3}$ | $7e^{-3}$ | $8e^{-6}$ |
| $\eta$ | CS $\Rightarrow$ CD | 0.50 | 0.10 | 0.50 | 0.55 |
| | CS $\Rightarrow$ IC | 0.30 | 0.50 | 0.55 | 0.40 |
| $\phi$ | CS $\Rightarrow$ CD | 0.65 | 0.55 | 0.35 | 0.35 |
| | CS $\Rightarrow$ IC | 0.60 | 0.60 | 0.30 | 0.55 |

(2022), we use the batch size of 8, 16, 16, and 8 for VOC, COCO, NUS, and CUB respectively. Consistent with Chen et al. (2024), we fix the learning rates to $1e^{-5}$ for VOC, COCO, and NUS, and to $5e^{-5}$ for CUB.

### C.4 Hyperparameters

We optimize the hyperparameters for each method and dataset by maximizing the validation MAP. Table 4 lists the final choices obtained by the process described below for all of our model variants. For all of the comparison methods, we use the same validation protocol and candidate sets reported in the original publications or their associated code-bases; we do not replicate the descriptions here.

First, we note that our code uses slightly different parameterization than the main paper: The weights in Equation 4 and 5 are re-parameterized as

$$\alpha = \frac{\gamma}{\eta^2}, \tag{6}$$

$$\beta = \left(\frac{\eta}{\phi}\right)^2, \tag{7}$$

and the validation is conducted in the space of $\gamma, \eta$, and $\phi$. Instead of a full grid search, we use a staged approach to reduce the computational cost and overfitting to the validation data:

- We tuned the $\lambda$ parameter of CS within the set $\{0.01, 0.1, 0.15, 0.2, 0.3, 0.35\}$. This also provides $e_w$ (the number of CS initialization epochs for CS$\Rightarrow$ CD and CS$\Rightarrow$ IC, see Section 4.5) for each dataset without requiring separate runs to tune this parameter.

- After determining $\lambda$, we search for $a$ in $\{1, 2, 3, 4\}$ for all datasets.

- For CS $\Rightarrow$ CD and CS $\Rightarrow$ IC, we perform a grid search of 64 configurations ($8 \times 8$) in $\{0.1, 0.2, 0.3, 0.35, 0.4, 0.45, 0.5, 0.55\}$ for $\eta$ and in $\{0.3, 0.35, 0.4, 0.45, 0.5, 0.55, 0.6, 0.65\}$ for $\phi$.

- The range of candidates considered for $\gamma$ depends on the case:

  - For VOC+COCO+NUS and CS $\Rightarrow$ IC, we search in $\{5e^{-3}, 6e^{-3}, 7e^{-3}, 8e^{-3}, 9e^{-3}\}$.
  - For VOC+COCO and CS $\Rightarrow$ CD, we search in $\{6e^{-5}, 7e^{-5}, 8e^{-5}, 9e^{-5}, 1e^{-4}\}$
  - For NUS and CS $\Rightarrow$ CD, we search in $\{2e^{-3}, 3e^{-3}, 4e^{-3}, 5e^{-3}\}$
  - For CUB and both methods, we search in $\{5e^{-6}, 6e^{-6}, 7e^{-6}, 8e^{-6}, 9e^{-6}\}$

Table 5: Test MAP for all our method variants and the naive baseline (AN) using the training data with a maximum of two labels per sample. The results for Single-positive from Table 1 are duplicated here for the ease of comparison.

| Method | VOC | COCO | NUS | CUB |
|---|---|---|---|---|
| Full label | $89.56 \pm 0.35$ | $76.43 \pm 0.09$ | $52.12 \pm 0.09$ | $32.73 \pm 0.20$ |
| **Single-positive** | | | | |
| AN | $85.51 \pm 0.41$ | $64.33 \pm 0.10$ | $42.68 \pm 0.09$ | $18.65 \pm 0.31$ |
| CS | $88.97 \pm 0.03$ | $71.35 \pm 0.10$ | $49.23 \pm 0.45$ | $21.56 \pm 0.08$ |
| CS $\Rightarrow$ CD | $89.10 \pm 0.17$ | $72.07 \pm 0.08$ | $50.26 \pm 0.08$ | $21.58 \pm 0.21$ |
| CS $\Rightarrow$ IC | $89.69 \pm 0.16$ | $73.45 \pm 0.12$ | $50.92 \pm 0.48$ | $21.73 \pm 0.35$ |
| **Two-positive** | | | | |
| AN | $89.07 \pm 0.17$ | $70.55 \pm 0.07$ | $47.78 \pm 0.29$ | $22.22 \pm 0.20$ |
| CS | $90.22 \pm 0.14$ | $73.55 \pm 0.08$ | $51.57 \pm 0.29$ | $22.76 \pm 0.38$ |
| CS $\Rightarrow$ CD | $90.11 \pm 0.07$ | $73.44 \pm 0.15$ | $52.38 \pm 0.14$ | $23.69 \pm 0.25$ |
| CS $\Rightarrow$ IC | $90.39 \pm 0.12$ | $75.50 \pm 0.03$ | $52.94 \pm 0.13$ | $23.94 \pm 0.09$ |

## D  Beyond SPMLL: General Partial Labeling Setups

Even though our main interest was in the SPMLL setup that requires the least amount of labeling effort and hence has the most potential for benefiting from the cardinality information, we provided the details for a more general formulation where $k_i^o$ labels are observed for the $i$th sample. To show that the method indeed works also in other setups, we conducted an experiment that is otherwise identical to the Experiment 1 but we now observe up to two positive labels for each sample, when available, while still considering the rest unobserved even if the sample did not have more than one true positive label. We re-use the hyperparameters (Table 4) from the main experiment to validate also the robustness of these choices, even across a change in the learning task.

Table 5 reports the MAP for the proposed method variants and the AN baseline, and repeats the results for the single-positive setup from Table 1 for convenience. The first, obvious, observation is that the MAP improves for every method and data set, as is to be expected since we know have more information. The main result, however, is that the proposed methods retain their advantage: We still have CS outperforming AN in all cases, and both CS $\Rightarrow$ IC is still the best method, with approximately the same improvement over CS in both scenarios.

