# OpenReview forum: "Single-positive Multi-label Learning with Label Cardinality"
_TMLR — Accepted by TMLR_

### Review · Reviewer_98sq · 2025-05-20

**Summary Of Contributions:**

This paper presents a framework of methods to use side information to improve the problem of SPMLL (single-positive multi-label learning). The side information discussed are mostly related to label cardinality (i.e. the number of labels per instance), with different granularities, instance cardinality (IC), cardinality statistics (CS), and cardinality distribution (CD). The authors propose a method for each setting, and introduce how to estimate IC with CS (i.e. trying to provide more side information) with an efficient bipartite matching algorithm. Experiments are performed on standard benchmark datasets,

**Audience:**

Yes

**Broader Impact Concerns:**

Not any.

**Claims And Evidence:**

No

**Requested Changes:**

1. Discussion/experiments about how the proposed method can work together with baseline methods.
2. Some better presentation forms for Figure 4.
3. Discussion on how results Table 1 and Table 4 are different.

**Strengths And Weaknesses:**

# Strengths
1. The paper is well-written and easy to follow. Technical backgrounds, problem formulation, and the definitions of side information are given clearly. Each technical method is introduced with sufficient intuitions, making it easy for readers to understand it.
2. The proposed methods are simple to implement.
3. This is my personal thought, but I believe that trying to use more side information from the data is a promising way to improve all kinds of machine learning tasks in general. This paper is one of such attempts and I appreciate it.
4. Experiment 2 is of practical value in that it answers the question of how accurate the estimations should be for the method to work.

# Weaknesses
1. Experimental results do not compare favorably against baseline methods. In Table 1, CS in general is not the best-performing method compared to baselines.
2. Extending from weakness 1, it is interesting if the proposed method can work together with some more advanced baselines. It would be good if the proposed method can work with baselines and further improve performance.
3. Figure 4 is slightly hard to interpret. I understand it is trying to show how accurate the label cardinality estimation is, but the visualization is not sufficiently clear. Maybe some other presentation forms can be used (e.g. distribution of estimation error).
4. It is slightly confusing about how the experimental settings differ between Table 1 and Table 4. The authors may provide some additional discussion.

---

> ### Author Response · Authors · 2025-06-23
> **Response to Reviewer 98sq**
>
> > Discussion/experiments about how the proposed method can work together with baseline methods.
>
> - We now extended the discussion to be more specific on how the ideas could be combined with other methods, making concrete link to both pseudo-labeling and active learning, but implementing and evaluating these hypothetical methods remains out of scope of the current paper.
>
> ***
>
> > Some better presentation forms for Figure 4.
>
> - Figure 10 in Appendix now includes an alternative visualization as a box-plot. We kept the original figure in the main paper as we believe it better communicates the full distribution, but agree that a more familiar box-plot may be useful for many readers.
>
> ***
>
> > Discussion on how results Table 1 and Table 4 are different.
>
> - Table 4 reports the hyperparameter values used for the reported results in Table 1. Perhaps you meant to write "Table 1 and 2" instead? In that case, the difference is in the validation information available for tuning the hyperparameters.
>
> - In Table 1, we assume validation data where all instances are *fully labeled*. All previous SPMLL methods use such validation data, and hence we need to use the same setup for the main comparison.
>
> - In Table 2, we make the more realistic assumption that we only observe a single positive label also for the validation samples. This naturally decreases the accuracy, but Table 2 confirms our method works also in this realistic setup and does not require access to fully labeled validation data.

---

### Review · Reviewer_tbHe · 2025-05-22

**Summary Of Contributions:**

This paper investigates the problem of single-positive multi-label learning (SPMLL), where each training instance is annotated with only one positive label. The authors propose a unified framework that leverages label cardinality information at multiple levels of granularity—from coarse global statistics to fine-grained instance-level estimates—to improve learning in this weakly supervised setting. The core contributions include (1) a cardinality-aware loss that aligns the predictions of unobserved labels with statistical priors, (2) a joint loss that enforces consistency between model outputs and estimated cardinalities, and (3) a bipartite matching algorithm designed to infer per-instance label cardinality from model predictions and global label statistics. The experimental results on benchmark datasets such as NUS-WIDE and CUB demonstrate that the proposed method can narrow the gap between SPMLL and fully-supervised training, while maintaining robustness under noisy cardinality estimates.

**Audience:**

Yes

**Claims And Evidence:**

Yes

**Requested Changes:**

Please refer to weaknesses

**Strengths And Weaknesses:**

Strengths:

The paper is clearly written, making the technical content accessible and easy to follow.

The hierarchical treatment of cardinality information -- distinguishing between statistics, distributions, and instance-level estimates—offers a structured perspective that could motivate future work in both SPMLL and broader weakly supervised settings.

The authors provide extensive ablation studies and sensitivity analyses, which enhance the empirical credibility.


Weaknesses:

From a technical perspective, the novelty of the proposed method is somewhat limited. The approach mainly combines existing techniques with heuristic adaptations, and the loss function design is largely intuition-driven, lacking a deeper theoretical rationale for why or how label cardinality contributes to improved learning under the SPMLL setting.

On the experimental side, there are several issues that limit the strength of the results. First, although the authors say that their loss function can be used as a plug-in module, they do not show how it works when combined with other existing SPMLL methods. This makes it hard to judge how useful or general the method really is. Second, the experiments do not include some important recent SPMLL baselines, such as the method by Xie et al. (NeurIPS 2022), which makes the comparisons less convincing. Third, all experiments are done only in the single-positive label setting, without testing whether the method can work in more general cases like missing-label learning. Lastly, although the method does improve performance, the gains over existing methods are quite small. This raises concerns about how much practical benefit the method really provides.

---

> ### Author Response · Authors · 2025-06-23
> **Response to Reviewer tbHe**
>
> > The approach mainly combines existing techniques with heuristic adaptations, and the loss function design is largely intuition-driven, lacking a deeper theoretical rationale for why or how label cardinality contributes to improved learning under the SPMLL setting.
>
> - We would like to emphasize that (a) the concept of using label cardinality information to encourage predictions consisting with the cardinality is novel, and (b) the algorithm for estimating the instance-cardinalities is both novel and well-justified, with guaranteed global optimal solution for that sub-task. We now clarified this in the Introduction.
>
> ***
>
> > although the authors say that their loss function can be used as a plug-in module, they do not show how it works when combined with other existing SPMLL methods.
>
> - We now clarified the discussion in Section 6, explaining concretely how it could be combined with one example method (EM+APL by Zhou et al.). We agree that it would be interesting to see how well such hypothetical combinations would work in practice, but we need to leave it as future work.
>
> ***
>
> > the experiments do not include some important recent SPMLL baselines, such as the method by Xie et al. (NeurIPS 2022), which makes the comparisons less convincing.
>
> - Our experiment is about comparing the effect of the loss and the algorithm, and hence we need to use the same backend model for all methods. We chose ResNet-50 as the most commonly used backbone and intentionally avoided including complicated elements into the base model.
> - Xie et al. (2022), which we already cited in the first version as well, indeed achieves very good MAP for these benchmark tasks but does so by substantially changing the model itself. The MAP scores across the models are not at all comparable, which is best seen by a massive 8 points difference in MAP even for the AN baseline. To compare against their method, we would hence need to change *all* baseline methods, not just our proposed method, to include their new model component. We believe this would reduce the value of the main experiment, since it would then only show the value of the cardinality information in context of the specific model they proposed. Now we show it in the context of the most common model.
> - We now clarify this in Section 6 and we also re-phrased the abstract to explicitly mention that our empirical evaluation is in context of alternative methods that do not change the model itself.
>
> ***
>
> > all experiments are done only in the single-positive label setting, without testing whether the method can work in more general cases like missing-label learning.
>
> - This is a good point. We focused on SPMLL as the cardinality information is expected to be of the most value in that setup, but it is indeed useful to show that the approach works also in other setups.
> We now added a new Experiment in Appendix D to demonstrate the approach in a setup with multiple (two) observed labels per sample. Table 5 shows using the cardinality information still helps over the baselines and confirms the accuracy improves when we observe more labels.

---

### Review · Reviewer_1tSR · 2025-06-09

**Summary Of Contributions:**

The paper introduces the use of label cardinality information—ranging from crude statistics to full instance-level cardinalities—as auxiliary supervision in single-positive multi-label learning (SPMLL), a setting where each training instance has only one observed positive label. This is a new setting and is interesting to study. They propose different loss functions and a heuristic workaround when only distribution of label cardinality is available. The proposed approach is tested on four SPMLL benchmark datasets and compared with multiple baselines. Results show that even with minimal additional information (e.g., mean cardinality), the method achieves competitive or superior performance, and further improvements are seen with more detailed cardinality knowledge.

**Audience:**

Yes

**Broader Impact Concerns:**

None.

**Claims And Evidence:**

Yes

**Requested Changes:**

Method Design Choices:

Some method design decisions are ad hoc. For example, the way thresholds are derived from summary statistics​  could be further justified.

Section 4.5: The two choices for $M_i$ (number of top-predicted labels) are stated without enough justification. Is there any theoretical or empirical analysis comparing them to other alternates?

Writing and Presentation:

Abstract: The term “simple method” could be clarified—simple in terms of implementation, computational cost, or sample efficiency?

Abstract: “State-of-the-art” should be qualified—within SPMLL or broader weakly supervised MLL?

Introduction: The sentence “a bipartite matching algorithm...” could benefit from some qualification. It currently implies this method is used universally without constraint, but it only applies under the CD assumption.

Related Work: It would be useful to clarify under what assumptions pseudo-labeling methods work better or worse than this method. Has any prior work combined pseudo-labeling with cardinality information? Can label cardinality help improve pseudo-label accuracy?

Method: In Section 4.5, the transition from LCS to LIC involves solving a cardinality estimation problem. The choice to use a greedy matching algorithm is motivated empirically, but could benefit from discussion on when the assumptions of monotonicity/sorting would break.

Additional Considerations

Is there any benefit in using the predicted label probabilities directly for pseudo-labeling in combination with cardinality constraints (e.g., sampling top-k based on cardinality estimates)?

Any potential use in active learning? For example, could estimated cardinalities guide which samples to annotate further?

**Strengths And Weaknesses:**

Strengths:

*Relevance* The problem is highly relevant in practical applications where labeling data comprehensively is resource-intensive. In many real-world scenarios, annotators can only provide partial label information—typically a single positive label per instance—despite the true class membership being multi-label. The setup reflects a practical relaxation of annotation assumptions.

*Simple Yet Effective Setup*: The authors present a minimal modification to the existing SPMLL pipeline by incorporating cardinality information. Despite the simplicity, the approach is shown to yield strong empirical performance, suggesting robustness and ease of adoption.

*Mathematical Motivation*: The paper provides a clear and structured mathematical formulation of the problem. Different forms of side information on cardinality (IC, CD, CS) are formally defined, and corresponding loss functions are derived with justification. The treatment helps clarify the assumptions and goals, aiding reproducibility and further development.

*Clear Writing* The manuscript is generally well-written and easy to follow. Notably, Section 4 begins with an overview paragraph, which is helpful for navigating the subsequent content—similar summaries for other sections would be useful for a reader and the authors are encouraged to add them. The figure captions are informative, and I appreciate the effort to remain to the point while conveying necessary detail is appreciated.

*Comprehensive Experimental Evaluation* The authors conduct a broad set of experiments across four datasets and multiple baselines. The analysis covers robustness to estimation error, ablations on estimation techniques, and sensitivity analyses.

Weaknesses:

1. *Baseline (Naive) Approach*: The “Assumed Negative” (AN) baseline is perhaps too weak to be informative. Given the known limitations of treating unknown labels as negatives, including it primarily serves to highlight the inadequacy of this assumption. Additionally, the choice of quadratic loss for Lu (in Equation 3) is not fully motivated. Are there alternatives considered (e.g. hinge, BCE)? Why is quadratic preferred?


[IMPORTANT CONCERNS]

2. *Cardinality Distribution Assumption*: The paper assumes access to the label cardinality distribution (CD) in some settings. It is unclear how this distribution would be obtained in practice. If estimated, what are its statistical properties, are there any assumptions you make, can it be heavy-tailed?

3. *Learning Cardinalities via Loss*: The claim that the model’s predictions carry information about cardinality under the proposed loss functions (especially LCS and LIC) could use more theoretical support. I understand the bipartite matching would work and optimize the given function, but why do the f_i contain enough information Why does the proposed loss structure lead to meaningful cardinality estimates? Is there any inductive bias introduced that aligns the model outputs with cardinality? This is particularly relevant when no instance-level supervision on cardinality is provided.

---

> ### Author Response · Authors · 2025-06-23
> **Response to Reviewer 1tSR -- Part 1**
>
> > the choice of quadratic loss for $L_u$ (in Equation 3) is not fully motivated. Are there alternatives considered (e.g. hinge, BCE)? Why is quadratic preferred?
>
> - The purpose of the loss is merely to make all probabilities within the interval equally likely. We chose a quadratic penalty for violations on the ground of simplicity (quadratic penalties are familiar for all readers), but broad range of monotonically increasing penalty could be used as well. We now mention this in the paper.
>
> ***
>
> > The paper assumes access to the label cardinality distribution (CD) in some settings. It is unclear how this distribution would be obtained in practice. If estimated, what are its statistical properties, are there any assumptions you make, can it be heavy-tailed?
>
> - Experiment 3 empirically demonstrates the method in cases where CS is estimated from samples of varying sizes. We did not analyze the estimator's properties, but showed that there is natural link between the estimation accuracy and MAP (see also Appendix B.2 and Figure 8). This is a standard discrete density estimation problem and should behave well outside of pathological cases, such as outlier samples with extremely high cardinality, and we now mention that robust estimators for the max cardinality could help in such cases.
> - We did not address the issue of how the cardinalities for individual samples would be annotated, but note that e.g. in audio applications the cardinality would correspond to polyphony that could be easier to annotate than a full label vector.
>
> ***
>
> > The claim that the model’s predictions carry information about cardinality under the proposed loss functions could use more theoretical support. I understand the bipartite matching would work and optimize the given function, but why do the $f_i$ contain enough information Why does the proposed loss structure lead to meaningful cardinality estimates? Is there any inductive bias introduced that aligns the model outputs with cardinality? This is particularly relevant when no instance-level supervision on cardinality is provided.
>
> - An informal justification is the same as it is for partial label algorithms in general: We expect $f_i$ to carry information about the missing labels since the network has been trained to recognize features corresponding to that class due to the other training examples. For example Kim et al. [1] (LL-R+BoostLU in Table 1) illustrated this using class activation maps for missing labels, and the success of SPMLL algorithms in general tells the predictions carry information about the missing labels.
> - As of now, the accuracy of estimating the cardinalities remains an empirical question. We designed both the CS loss and the cardinality regularizer to encourage $f_i$ to carry information about the cardinality, and demonstrated that we indeed can infer the instance cardinalities accurately enough to improve the overall MAP. Future analysis on how the quality of $f_i$ correlates with the cardinality estimates and ultimately the MAP would be interesting.
>
> ***
>
> > Some method design decisions are ad hoc. For example, the way thresholds are derived from summary statistics could be further justified.
>
> - The thresholds are justified in the two last paragraphs on page 5. In the end, these are heuristics rather than exact derivations and our main goal was just to ensure that they would not need to be considered as hyperparameters but could be set automatically.
>
> ***
>
> > Section 4.5: The two choices for $M_i$(number of top-predicted labels) are stated without enough justification. Is there any theoretical or empirical analysis comparing them to other alternates?
>
> - We view $M_i$ as a technical parameter that is not particularly important but just needs to be sufficiently large. We already showed in Appendix (Section B.3, Figure 9) that $M_i \approx k_i$ would not work but other than that the method is not sensitive to the choice. For the data sets considered here, we could also use $M_i = k_{\text{max}} -1$.
>
> ***
>
> > Abstract: The term “simple method” could be clarified—simple in terms of implementation, computational cost, or sample efficiency?
>
> - We re-phrased this to clarify we meant conceptual or implementation simplicity.
>
> ***
>
> > Abstract: “State-of-the-art” should be qualified—within SPMLL or broader weakly supervised MLL?
>
> - We meant within SPMLL and specifically within methods that use the same underlying architecture, matching our empirical comparisons. We now clarified this.

---

> ### Author Response · Authors · 2025-06-23
> **Response to Reviewer 1tSR -- Part 2**
>
> > Introduction: The sentence “a bipartite matching algorithm...” could benefit from some qualification. It currently implies this method is used universally without constraint, but it only applies under the CD assumption.
>
> - Our goal was to say we introduce three separate components that could also be used in isolation, and the sentence mentioning bipartite matching indicated it is proposed for the specific task of learning instance cardinalities. That said, we now clarified the writing to make it explicit that the components require different kind of assumptions on available data.
>
> ***
>
> > Related Work: It would be useful to clarify under what assumptions pseudo-labeling methods work better or worse than this method. Has any prior work combined pseudo-labeling with cardinality information? Can label cardinality help improve pseudo-label accuracy?
>
> - We are not aware of any pseudo-labeling methods considering label cardinalities, but do see it as a promising future avenue. Like you said in a later comment, incorporating cardinality estimates as constraints for pseudo-labeling would be intuitively a very good idea. We now expanded the discussion to explicitly mention this opportunity, while remarking that the empirical performance would need to be validated before making claims on how well they work.
>
> ***
>
> > Method: In Section 4.5, the transition from $L_{CS}$ to $L_{IC}$ involves solving a cardinality estimation problem. The choice to use a greedy matching algorithm is motivated empirically, but could benefit from discussion on when the assumptions of monotonicity/sorting would break.
>
> - Note that we prove in Appendix A that this greedy algorithm provides a globally optimal solution for the specified assignment problem. We now see how the description can be misread as referring to a *greedy approximation*, and hence clarified the writing to avoid giving this impression.
>
> ***
>
> > Any potential use in active learning? For example, could estimated cardinalities guide which samples to annotate further?
>
> - We had not thought about this, but indeed this could be a possible use-case. We now mention it briefly in Discussion as one possible direction for future work.

---

> > ### Comment · Reviewer_1tSR · 2025-08-11
> >
> > I thank the authors for their comments, my concerns were indeed resolved and I had submitted my recommendation a while back, I should have communicated this to you. You should check with the other reviewers if their concerns were addressed and if they have submitted a recommendation.

---

### Review · Reviewer_MrUA · 2025-06-10

**Summary Of Contributions:**

The authors address the problem of Single Positive Multilabel Learning (SPML) by leveraging additional information on label cardinality. Their primary contribution lies in integrating label cardinality information at different levels of granularity into the optimization problem's design. Specifically, they explore three forms of label cardinality information: instance cardinality (IC), cardinality distribution (CD), and cardinality statistics (CS). The paper introduces distinct methods for embedding this cardinality information directly into the optimization objective and evaluates the proposed approach through numerical experiments on benchmark datasets, demonstrating its effectiveness and applicability.

**Audience:**

Yes

**Claims And Evidence:**

Yes

**Requested Changes:**

Please see the previous section.

Can the authors comment on the following question:
Is it possible to retain the objective functions in (1) as they are in standard methods while incorporating label cardinality information as constraints in the optimization problem? This approach could, in principle, allow leveraging the strengths of established SOTA methods while also integrating the additional insights provided by label cardinality information.

Minor comments:
It is better for the readability to use \citep when citations are not part of the sentence. For example:
"Numerous methods for learning multi-label classifiers from fully annotated data have been proposed (Ridnik et al., 2023; Zhang & Zhou, 2013; Parascandolo et al., 2016)..."
reads better than
"Numerous methods for learning multi-label classifiers from fully annotated data have been proposed Ridnik et al. (2023); Zhang & Zhou (2013); Parascandolo et al. (2016)..."

**Strengths And Weaknesses:**

**Strengths**
- The authors address the critical problem of SPML, recognizing that obtaining fully annotated data can be prohibitively expensive or challenging in certain applications.
- The proposed methods are intuitive and appear straightforward to implement, making them accessible for practical use.
- The methods exhibit a plug-and-play nature. Adding numerical experiments to validate this characteristic, by integrating these methods with other standard techniques and demonstrating performance improvements, would further strengthen the paper's contributions.

**Weaknesses**
- Many of the proposed approaches rely on intuition and lack theoretical guarantees. While this is not inherently a drawback if the methods demonstrate significant performance improvements over the state of the art (SOTA), the reported results show that performance is, at best, comparable to baseline methods.
- The use of a quadratic loss function in (3) is presented as intuitive, but the rationale for this choice is underexplored. It raises the question of whether alternative loss functions, for example, the absolute loss function, were considered. The authors should provide a more detailed justification for their choice, supported by empirical evidence where possible.
- The reliance on $k_{\max}$ and $k_e$ as representatives of label cardinality information may lack robustness. It seems plausible that a single outlier could skew these metrics, making the proposed approach vulnerable to datasets with such outliers. This raises concerns about the generalizability of the method across datasets with varying label distributions.
- Problem-specific hyperparameter tuning using grid search contradicts the plug-and-play nature of the proposed methods. This dependency on tuning undermines the claim of ease of integration.
- The matching algorithm described in Section 4.4 for estimating instance cardinalities is intriguing. However, it is unclear whether the current model predictions, $\mathbf{f}_i$, provide sufficient information to compute reliable scores.

---

> ### Author Response · Authors · 2025-06-23
> **Response to Reviewer MrUA -- Part 1**
>
> > The use of a quadratic loss function in (3) is presented as intuitive, but the rationale for this choice is underexplored. It raises the question of whether alternative loss functions, for example, the absolute loss function, were considered. The authors should provide a more detailed justification for their choice, supported by empirical evidence where possible.
>
> - The purpose of the loss is merely to make all probabilities within the interval equally likely. We chose a quadratic penalty for violations on the ground of simplicity (quadratic penalties are familiar for all readers), but broad range of monotonically increasing penalty could be used as well. We have clarified this in the paper.
>
> ***
>
> > The reliance on $k_{\text{max}}$ and $k_{e}$ as representatives of label cardinality information may lack robustness. It seems plausible that a single outlier could skew these metrics, making the proposed approach vulnerable to datasets with such outliers. This raises concerns about the generalizability of the method across datasets with varying label distributions.
>
> - The mean $k_e$ is essentially the easiest quantity to estimate and in any case a heuristic approximation that could be replaced by any small constant, and hence estimator error of $k_e$ is of no concern.
> - The maximum is indeed harder to estimate, but we show in Figure 7 (in Appendix) that the method is extremely robust even for errors of a factor of two. By replacing the estimator with a robust one (e.g. 95\% quantile estimator) we would get a stable estimate that may have a small bias but according to Figure 7 such a bias will not matter. We now clarify this both when introducing the cardinality statistics and  in Appendix B.1 when discussing the sensitivity analysis of $k_{\max}$.
>
> ***
>
> > Problem-specific hyperparameter tuning using grid search contradicts the plug-and-play nature of the proposed methods. This dependency on tuning undermines the claim of ease of integration.
>
> - We indeed need to optimize the hyperparameters, but this is the case for all deep learning methods. The comment about plug-and-play means ability to replace components in previously proposed methods with updated variants, especially the loss for the unobserved labels ($L_\text{CS}$). Note that the CS loss has *fewer* hyperparameters than most SPMLL methods, just the rate of penalizing around the flat area, since the other parameters are determined by the cardinality statistics.
> - In practice the method is relatively robust to the choices. To demonstrate this, the new experiment in Appendix D (Table 5) was done with the hyperparameter values obtained for the results reported in Table 1. That is, we did not do *any* extra tuning and the results are still good, even though here even the task itself has changed.
>
> ***
>
> > The matching algorithm described in Section 4.4 for estimating instance cardinalities is intriguing. However, it is unclear whether the current model predictions, $\mathbf{f}_i$, provide sufficient information to compute reliable scores.
>
> - An informal justification is the same as it is for partial label algorithms in general: We expect $f_i$ to carry information about the missing labels since the network has been trained to recognize features corresponding to that class due to the other training examples. For example Kim et al. [1] (LL-R+BoostLU in Table 1) illustrated this using class activation maps for missing labels, and the success of SPMLL algorithms in general tells the predictions carry information about the missing labels.
> - As of now, the accuracy of estimating the cardinalities remains an empirical question. We designed both the CS loss and the cardinality regularizer to encourage $f_i$ to carry information about the cardinality, and demonstrated that we indeed can infer the instance cardinalities accurately enough to improve the overall MAP. Future analysis on how the quality of $f_i$ correlates with the cardinality estimates and ultimately the MAP would be interesting.
>
> ***
>
> [1] Kim, Youngwook, et al. "Bridging the gap between model explanations in partially annotated multi-label classification." Proceedings of the IEEE/CVF Conference on Computer Vision and Pattern Recognition. 2023.

---

> ### Author Response · Authors · 2025-06-23
> **Response to Reviewer MrUA -- Part 2**
>
> > Is it possible to retain the objective functions in (1) as they are in standard methods while incorporating label cardinality information as constraints in the optimization problem? This approach could, in principle, allow leveraging the strengths of established SOTA methods while also integrating the additional insights provided by label cardinality information.
>
> - This is technically possible: The regularizer in Eq. 3 is additive and could be combined with any loss, and the instance cardinality estimation algorithm could be applied on top of any model predictions. This is what we meant when saying the components are plug-and-play, and we now clarified it in Section 6.
> - The accuracy of such a combination remains an open question. We specifically designed the CS loss to encourage $f_i$ being more informative, preventing collapse to constant values as in some established methods, but it could indeed be that some SOTA models could otherwise produce so good $f_i$ that we could still infer the instance cardinalities well and could use the information to regularize them further. This is an interesting path for future work.
>
> ***
>
> > It is better for the readability to use \citep when citations are not part of the sentence.
>
> - Indeed; we apologize for the mistake and fixed it.

---

### Author Response · Authors · 2025-06-09
**General comments**

We appreciate the reviewers’ comments and have carefully addressed each of them. We are currently awaiting the third reviewer’s feedback before submitting our responses along with the revised manuscript.

---

### Author Response · Authors · 2025-06-23

Thank you for your thoughtful review. We have uploaded a revised version of the manuscript, with all changes highlighted in blue. Below, we have provided detailed responses to each of your comments.

---

### Decision · Action_Editor_YsWu · 2025-08-12

**Recommendation:** Accept as is

**Audience:**

Yes

**Audience Explanation:**

The authors propose to address an interesting problem: how to leverage the label cardinality information for single-positive multi-label learning, which has many real-world applications. Reviwers also mentioned that the algorithm is accessible for practical use.

**Claims And Evidence:**

Yes

**Claims Explanation:**

The paper is clearly written with efficient experimental support. The research problem, motivation, and effective methodology are all clear.